



# Accelerating research through community open source software for a standardized file format to improve process representation in numerical weather prediction models

Johanna Tjernström[1,2], Michael Gallagher[3,4], Jareth Holt[5], Gunilla Svensson[5,6], Matthew D. Shupe[3,4], Jonathan J. Day[7], Lara Ferrighi[2], Siri Jodha Khalsa[8,3], Leslie M. Hartten[3,9], Ewan O'Connor[10], Zen Mariani[11], and Øystein Godøy[2]

[1]Swedish Meteorological and Hydrological institute, Norrköping, Sweden
[2]Norwegian Meteorological institute, Oslo, Norway
[3]Cooperative Institute for Research in Environmental Sciences, University of Colorado, Boulder, CO
[4]Physical Sciences Laboratory, NOAA; Boulder, CO
[5]Department of Meteorology and Bolin Centre for Climate research, Stockholm University, Stockholm, Sweden
[6]KTH Royal Institute of Technology, Department of Engineering Mechanics, Stockholm, Sweden
[7]European Centre for Medium-Range Weather Forecasts, Reading, United Kingdom
[8]National Snow and Ice Data Center, University of Colorado, Boulder, CO
[9]Physical Science Laboratory, National Oceanic and Atmospheric Administration, Boulder, CO
[10]Finnish Meteorological Institute, Helsinki, Finland
[11]Meteorological Research Division, Environment and Climate Change Canada, Toronto, Canada

**Correspondence:** Johanna Tjernström (johanna.tjernstrom@smhi.se)

**Abstract.** Improvements in process representation in numerical weather prediction (NWP) models requires informed collaboration between scientists making research-grade observations and scientist developing state-of-the-art NWP models. As a result, progress in model quality relies heavily on the ability to efficiently evaluate and reliably reconcile these two sources of information. To facilitate such progress, with focus on enhanced model skill in polar regions, the Year of Polar Prediction site Model Intercomparison Project (YOPPsiteMIP) community defined the Merged Data File (MDF) format. The file format is designed for high temporal and spatial resolution data for direct comparison between observations and model output to assess parameterized processes under various conditions. A broad overview of the MDF format is provided along with supporting use-cases defined by the research community, and present a set of free, open-source, computational tools for creating and utilizing this standardized format. Two free open source Python packages are discussed: 1) "The MDF toolkit", a data processing library for the creation of standardized datasets, and 2) "MDF visualization", a set of Python codes in notebook format that accelerate model evaluation and climate process research utilizing the MDF format. The benefits of such tools that may help unite diverse groups of researchers through a common data-format language are also discussed.



## 1  Introduction

With the increasingly data driven nature of modern day science, the architectures surrounding data creation, formatting and dissemination have become vital for scientific progress. In the current digital ecosystem the need to increase the uptake of data has created a burden on the users, as the data sources are often varied and formats are disparate. The state and quality of the data also varies, dependent on the policies and practices used by the data creators, and relevant documentation is often scarce. This serves as a hindrance towards collaboration and data use across research communities, which slows the progress of discovery. When standards do exist, they tend to be set on either a very high level (international standards) or on an individual basis (researcher) either by the data producers or data providers, thus making the task of finding, unifying and using the data cumbersome and time consuming. It also complicates inter-community collaborations, even if those communities exist within the same research discipline. In an attempt to push data creators and archives towards a more unified standard, the FAIR principles were outlined, focusing on making data Findable, Accessible, Interoperable and Reusable. (Wilkinson et al., 2016).

FAIR principles take into consideration the hurdles that data users, be they researchers or machines, face when trying to conduct research, and outlines a set of criteria that strive to improve the digital ecosystem surrounding research data, across disciplines. Through collaboration between academic groups and private stakeholders, guidelines were set for metadata conventions, proper referencing, how to make the data easier to find and access and easily usable for both researchers and machines. Thus the principles set up a framework towards ensuring that the data created becomes an asset to the wider community. However, they do not direct the specific details that makes it possible to work efficiently across communities, even if they are close subject-wise and are driven by the same overarching research questions. There is a need to get more specific to target specific cross-community research questions. Here, we present and discuss identified steps and measures, on both the technical and the human side, needed to to make progress with the aim to facilitate a more rapid progress of discovery. The methodology is general and can be applied on many research questions, although, our particular experience comes with the specific target to improve process representation in numerical weather prediction (NWP) models. Similar initiatives are being taken by multiple other communities as well e.g. (Neelin et al., 2023; Weil et al., 2023).

NWP models, require observational data for different purposes and at different stages. The most obvious one is through the data assimilation step to find the initial state of the atmosphere from which the forward integration can begin. Observational data is also needed for verification to assess to useful the forecast is. For both these purposes, international standards are set for data by the WMO. The purpose here is the usage of research-grade observations used to evaluate the process representation of small-scale processes (smaller than resolved by the computational grid) that affects the forecasts weather parameters. In this case the different data formatting and archival practices used in the observational and modelling communities become a hurdle. To evaluate an NWP model, the researcher needs to access observational data for multiple geographic locations, depending on the geographic coverage of the model. These different observational sites are run by different institutions in different countries and have different internal formats, varying levels of documentation and make the data available in different ways. Once that data has been gathered, it needs to be formatted in a way where it can be easily used alongside the data from the model, which has its own format. This process is further complicated if the number of models being evaluated is increased. This is



an overwhelming task, especially, for example, in the case of a global model developer who aims to evaluate model process representation in all climate zones across the globe.

The Year of Polar Prediction site Model intercomparison project (YOPPsiteMIP) was started with the aim of improving the
representation of parameterized processes such as stably stratified boundary layers and mixed-phase clouds (Uttal et al., 2023; Day et al., 2023). The project was initiated by the process task team of the World Weather Research Program (WWRP) Polar Prediction Project (Jung et al., 2016). Model improvement relies on proper understanding of the model issues and possible compensating biases that needs to be tackled simultaneously. Therefore, the first step is to evaluate and to assess the model performance. The project was designed to target small-scale processes in the atmospheric boundary layer and interaction
with the surface that are more difficult to represent in polar regions. A number of locations where suitable observations are made were identified (Mariani et al., 2024) and a number of NWP centers were invited to contribute with high-resolution model output. Seven numerical weather prediction models at seven observational sites in the arctic participated in the first YOPPsiteMIP study Day et al. (2023). A need for a common file format for efficiently sharing and archiving the model and observational data was identified and a collaborative community effort involving scientists from the NWP model groups and
from supersite observers was initiated.

The Merged Data Format (MDF) was created to address the common hurdles faced when performing multi-model evaluation using in situ observations from multiple sites (Uttal et al., 2023), with the goal to reduce the effort required to standardize data sets, Figure 1. The MDF specification has many benefits, especially that it has been developed by the community with the goal to standardize the important aspects of data processing so that any data produced to the specification can be readily used for
intercomparison or evaluation. Considerations were made for variable names, variable types, time cadences, vertical binning of profile information, radiosonde data processing, model grid-point selection, and more with the express goal of creating research quality data sets for model intercomparison and improvement (Uttal et al., 2023). This paper deals primarily with versions 1.0-1.3 of the MDF format, although the idea behind the format, as well as the key characteristics will remain applicable to future versions.



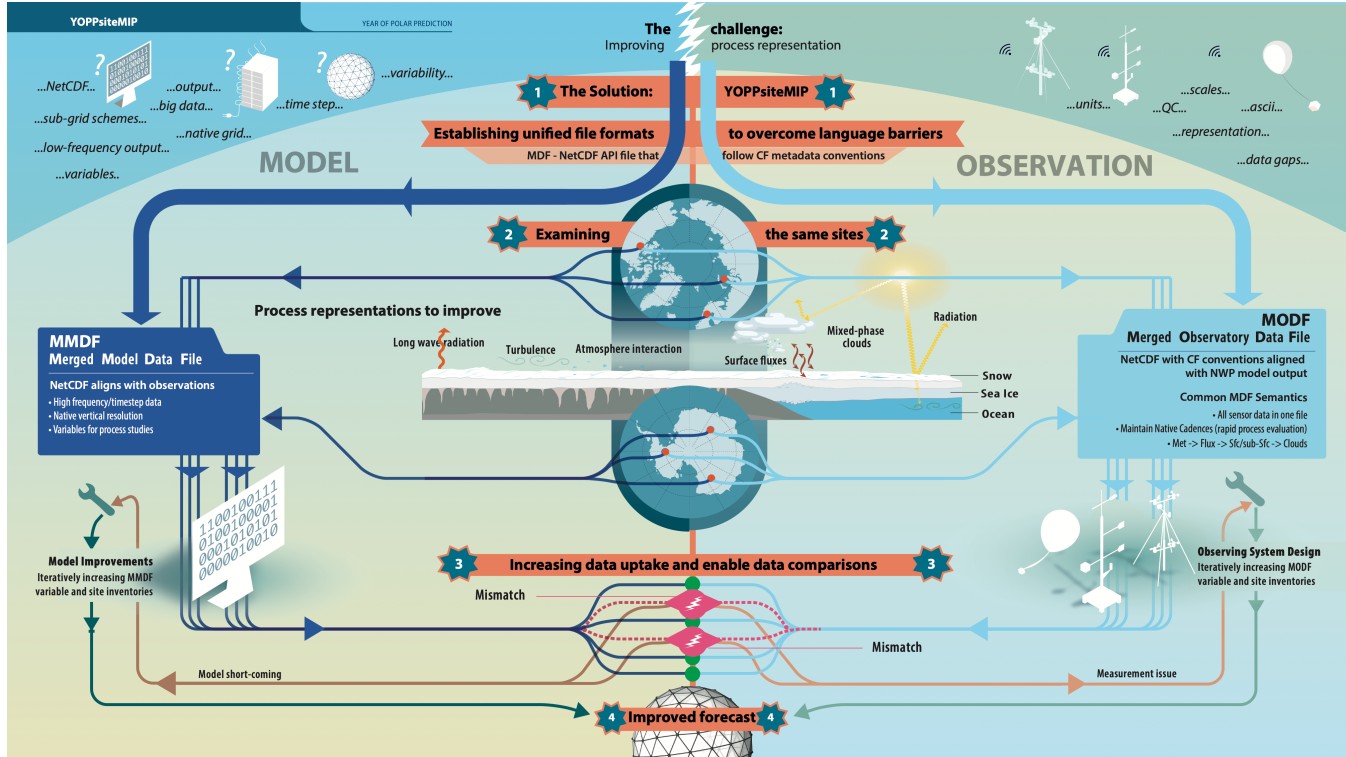

**Figure 1.** A schematic view showing the workflow of YOPPsiteMIP and how "Merged Data Files" accelerate models evaluation and improvement. The utility of MDFs is in having a shared common language for data from both in situ observations and model outputs, facilitating rapid iterative research. Image by Martin Kuenstig.

However, the existence of a standard format alone is not necessarily enough to keep files consistent. This was evident during the creation of the files for YOPPsiteMIP, where the format was given but no common tools were used. Thus the resulting files contain numerous differing format interpretations. For example, the sonde files contain three different combinations of dimension variable names; "time" and "height_sonde" , "time_release_sonde" and "hgt_sonde", "time_nominal" and "hgt_sonde". This type of issue is also present within the MMDFs, which contain two separate solutions for handling multiple gridpoints, neither of which match the way that MODFs handle multiple subsites. This clearly shows the need for a standardized file creation toolkit in order maintain the standard of the MDF files being created. It must be noted, that some of these discrepancies in the YOPPsiteMIP files are a result of the files being created alongside the creation of the format. However, moving forward with the usage of a common toolkit to create files will allow the submitted files to be closer to a correct standard when they are submitted, which will limit the amount of correction work needed.

The MDF Toolkit (Gallagher and Holt, 2023) is a free and open source (GPLv3) Python code that provides key functionality to process data for the creation of MDFs, saving users time and effort in producing their files for consumption. These key functions are: the automatic merging and conversion of timeseries data to MDF standard cadences, the proper metadata assignment according to research community standards, technical implementations of vertical profile timeseries and radiosonde





data, and finally the ability to evaluate and "check" the validity of MDFs created or modified by other tools. As such, the
MDF toolkit is tailored to the needs of researchers working with in situ observations and/or corresponding output data from
numerical models i.e. supersites and NWP forecast data in the case of YOPPsiteMIP. While a user of the MDF toolkit is still
responsible for the ingest of these types of data, when properly sourced the toolkit ensures that the appropriate community
standardized post-processing is applied, that the necessary metadata is present, and that the format of the resulting file written
to disk matches the MDF specification. These files can then easily be used by any other researchers in the siteMIP community.
The MDF visualization toolkit (Tjernström, 2024) provides web applications containing figures for a set of use-cases initially
outlined by those who have used the datasets for NWP evaluation during YOPPsiteMIP (Day et al., 2023). They are intended
as a starting point for data analysis, as tools that will facilitate the initial investigation by a potential a researcher, as well as an
aid to be used in education by lowering the threshold for analysing details in models as guided by observations.

## 2   The Merged Data File Format (MDF) specification

MDFs are formatted in the network Common Data Form (netCDF), a machine independent format for packaging scientific
data (Rew and Davis, 1990). netCDF abstractions allow for grouping, storage, and retrieval of multidimensional data such that
dissimilar types of data may coexist in a heterogeneous network without relying on formats specific to machines or applications.
In addition to this they allow for rich and complete syntactic and semantic descriptions of any desired metadata. The netCDF
format is also self describing, meaning that the file itself includes all information about its data. This makes netCDF files, when
data is encoded following set standards, easy to use and interpret without needing additional documentation or specific tools.
The MDF specification was created to make as efficient use as possible of the self describing nature of netCDF for the purpose
of facilitating model evaluation and intercomparison.

Although the goal of the MDF format is to be as similar as possible regardless of the origin of the data, some fundamental
differences between NWP and observational data cannot be avoided. For example, NWP forecasts for multiple days that start
at a regular interval with have overlapping time stamps, which needs to be catered for. Thus, beyond the criteria outlined for
standardization of data the MDF standard allows for some differences in the internal representation of different types of data.
To this end two sub-categorizations to the MDF file type were created as merged model data files ("MMDFs") and merged
observation data files ("MODFs"). This separation allows the user to capture specifics necessary to both observational data and
model data while still maintaining the underlying standard.
To further group similarly formatted types of data, the concept of "feature-types", used by the Climate and Forecast (CF)
conventions (Hassell et al., 2017), was added to the MDF standard. The MDF standard currently makes use of two feature-
types to create two main categories of data, timeSeries and timeSeriesProfile, Figure 2. For further specialization, a sub-type
has been added to the timeSeriesProfile feature-type, the timeSeriesProfileSonde.

The "timeSeries" feature-type represents data from a single "level"/height indexed simply by time, a common example
of data in the timeSeries category are near surface variables. The timeSeriesProfile feature-type represents a variable that is





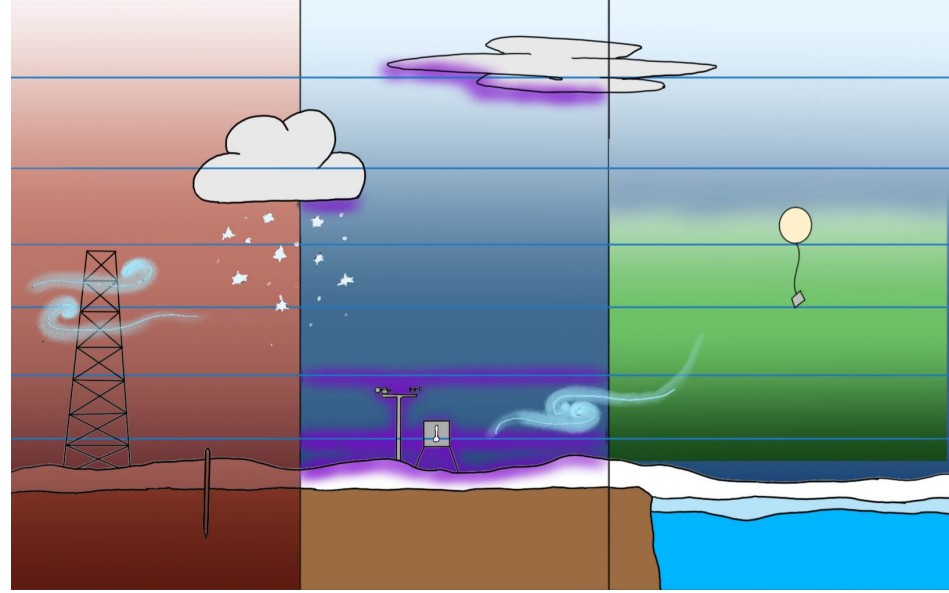

**Figure 2.** A visual representation of what variable types are encompassed by which feature types in YOPPsiteMIP, with timeSeriesProfile (multilevel) in red, timeSeries (single level) in purple and timeSeriesProfileSonde (sounding data) in green.

indexed by both height and time, for example tower data. The timeSeriesProfileSonde feature-type exists as a sub-type to timeSeriesProfile, tailored to the technical qualities of atmospheric sounding data.

For the YOPPsiteMIP MDFs the decision was made to allow multiple feature-types in the same file. The addition of multiple grid-points or different locations of instruments within one site remain encapsulated under the timeSeries and timeSeriesProfile
feature-types despite the addition of sub-site indexing.

The bulk of the MDF specification, and the basis for the feature-type categorizations, reside in the Hartten-Khalsa (H-K) table (Hartten and Khalsa, 2022). The H-K table builds on and extends the Climate and Forecast (CF) conventions (Hassell et al., 2017) and uses CMIP6 (Eyring et al., 2016) like naming conventions. It outlines the global attributes, temporal dimension variables, spatial and other dimensional variables, as well as six categories of geophysical variables. For each variable it lists
information about variable naming, a long name, a CF standard name or in cases where one does not yet exist a CF-like name is used, units, minimum recommended attributes and notes as well as if it is applicable to the MODFs, MMDFs or both. The H-K table exists in both a human readable format, pdf, as well as in the machine readable JavaScript Object Notation (json) format. It has been co-developed to reflect the needs of the users and the surrounding community and has been designed to have the capacity to grow and evolve as the user-base grows. For more in depth descriptions of the variable categories and the H-K
table's development process see (Uttal et al., 2023). The toolkits described in this section both rely heavily on the set standard of the H-K table. In turn their development, as well as the submission of files during and after YOPPsiteMIP have guided the development of the version 1.3 of the H-K table. The MDF toolkit can also be viewed as a tool to interface with and advance the H-K table, and aims to make it easier to apply the H-K table correctly when creating MDF files.





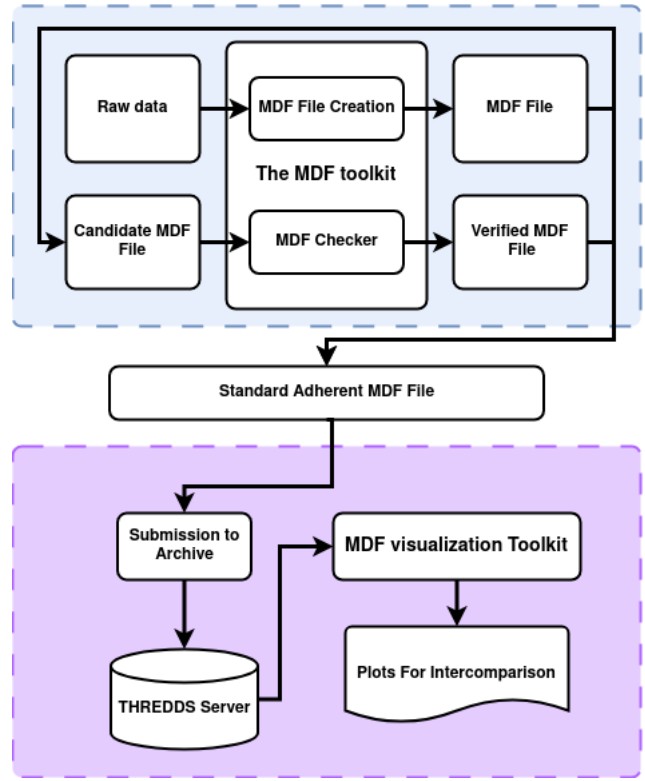

**Figure 3.** A high level overview of the MDF toolkit suite code ecosystem, with the MDF creation toolkit at the top and MDF visualization at the bottom, showing the data-flow from raw input data to MDF file, to archive and finally to the generation of plots for process based model evaluation.

## 3 The Merged Data File toolkit suite

The MDF toolkit suite contains two toolkits that are both aimed at facilitating the process of working with the MDF format and files, Figure 3. The MDF toolkit, consisting of a file creation tool and a checker aims to aid users in creating files, as well as checking the standard adherence of existing files. The MDF visualization toolkit aims to provide an easy way to interact with datasets created using the MDF standard, as well as give an insight into the process based model evaluation performed during YOPPsiteMIP.

### 3.1 "MDF Toolkit" — free software for creating standardized files

The MDF toolkit is a Python data-processing framework that facilitates the standardization of research datasets. It is based on the common Python libraries used in data science and in earth science research, primarily NumPy (Harris et al., 2020), Scipy (Virtanen et al., 2020), Pandas (The Pandas development team, 2020), and xarray (Hoyer and Hamman, 2017). This means that



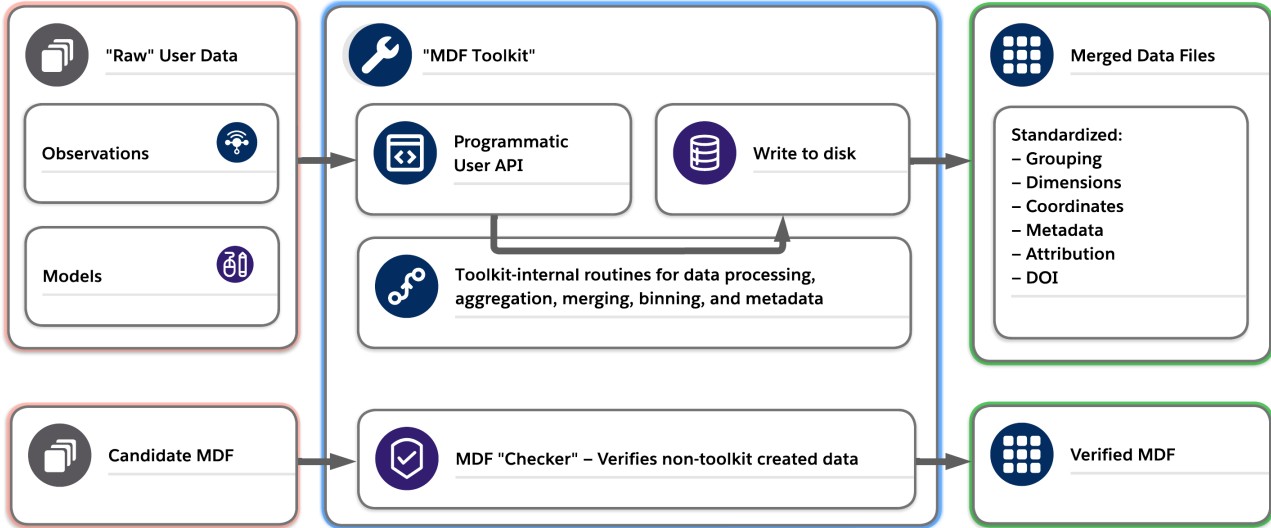

**Figure 4.** A high level overview of the flow of data through the MDF toolkit, starting as "raw" user data ingested into the toolkit, the internal functionality for data processing and finally being output as an MDF file. The input data can also be a candidate MDF file, which passes through the MDF checker, which outputs a report on where the candidate file follows the MDF specification and suggests necessary changes.

toolkit code can be directly plugged into a user's processing pipeline and MDFs can easily become part of standard routines

for observers and modelers alike provided they have a basic knowledge of Python.

The MDF toolkit consists of two main parts, the file creation functionality and the MDF checker, Figure 4. The user provides the file creator with timestamped data for the resulting MDF. Then based on the provided data, the toolkit runs standardized processing routines for common tasks such as vertical binning, alignment of time-series, metadata assignment, and more. The implementation details of specific processing routines, such as the vertical binning of profile variables, were determined by

an interdisciplinary team of modelers and observers convened for the siteMIP project, expressly for model improvement. The MDF "checker" provides the ability to validate and verify externally created MDF files. The user provides it with a candidate MDF and it checks the contents against the latest, or a specified, version of the H-K table and provides a report with deviations and suggested changes. Using the checker to guide file creation users do not need to use the creation tools, although it is encouraged. It is also encouraged to use the checker even if one has used the creation toolkit.

**3.1.1 Code and implementation**

The toolkit code for creating MDF files is implemented as a class interface, with a core set of reusable class member functions that maintain and process data according to the specification. This allows the code to be organized and reusable, and for the user-provided data to be contained within the state of the class object. In this way, a user of the toolkit creates their own MDF instance and then provides their data by calling the appropriate internal functions. Once the data has been provided, the





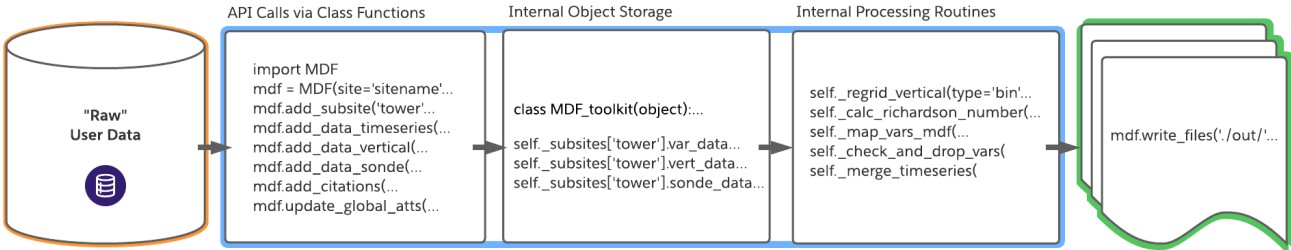

**Figure 5.** An example code path showing internal storage and functions that user data ingested by the toolkit follows. The API calls are the portion of the code that a user will provide data with. The toolkit uses a class-oriented storage design where the user data provided is ingested, processed, and stored internally until all processing routines have finished and write_files is called.

MDF class uses the data, metadata, and other information provided to call the proper standardized internal routines and, when requested, write the processed data and metadata to disk.

Figure 5 shows some of the common class functions and internal storage from common code paths used when creating MDFs. The "API calls" box shows the class functions that a user will call directly from their code while the "Internal Object Storage" and "Internal Processing Routines" boxes show some of the toolkit-internal code pieces that are implicitly being called

when creating the MDF. As such, the user simply provides the appropriate data and the internal toolkit provides the functional structure for the data standardization tasks. Following the common wisdom that good community code is much more often read than written, the toolkit is internally documented by means of Python docstrings and there is a strong emphasis on code clarity, readability (Hofmeister et al., 2017), and reusability.

### 3.1.2    "Checking" pre-existing or modified MDFs for conformance

While it is encouraged to use the toolkit to create MDFs there exists cases where files are created or modified in an ad-hoc manner based on pre-existing data processing routines. For this reason, and for the reason that many of the YOPPsiteMIP MDFs have been created by the data creators' own software, the toolkit includes a file checking functionality. The MDF checker takes a candidate MDF file and gives feedback on whether the file meets the proper MDF specification or the ways in which it does not. The ability to verify that files conform to the specification means that data archives and researchers are able to include

and use non-toolkit files, thus lowering the barrier for participation. The checker can be used programmatically by writing and running Python code to open the file, or run as a binary at the command prompt of your operating system.

### 3.2    "MDF Visualization" — website and codes for plotting and further analysis

The purpose of the visualization toolkit is dual. Firstly it was developed with the goal to serve as a web application alongside the YOPP data portal[1] to provide quick-looks for perspective dataset users who want to investigate the data further without

---

[1]https://yopp.met.no/



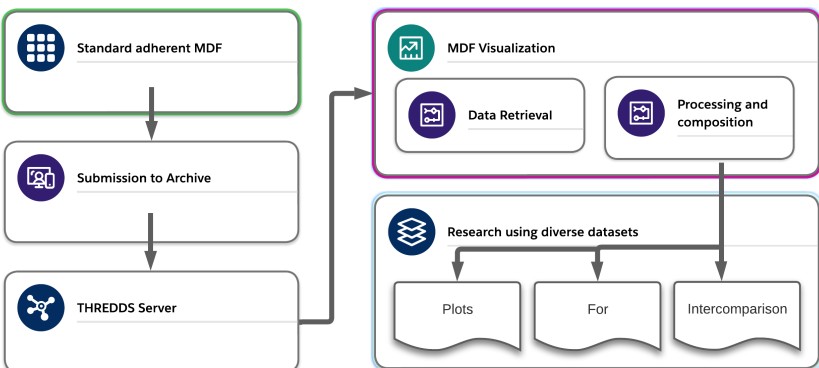

**Figure 6.** A high level overview of the functionality of the MDF Visualization toolkit as well as how it fits together with archived MDF standard data.

developing own software. Secondly it can function as stand-alone notebooks that can be used for educational purposes or as a starting point to lower the threshold to start developing their own code. The code has been made available via git, but has not yet become available as a web application for the portal. While the visualization toolkit was designed for process based evaluation of NWP models, many of the use-cases and the data handling functions can be used for other purposes.

### 3.2.1 System details and overview

The MDF visualization toolkit consists of a set of jupyter notebooks (Kluyver et al., 2016) spanning a set of 7 use cases assembled by the YOPPsiteMIP team from diagnostics for process based NWP evaluation done during the project (Day et al., 2023; Tjernström et al., 2021) Table 1. The system design is largely based around the modular nature of jupyter notebooks. Each use-case has its own notebook that contains information and functionalities specific to that use-case. Functionalities that are used in multiple use-cases, such as the functions available for fetching and formatting the data, as well as some common 190 calculations, exist in separate imported .py files. This allows for code reuse between the use-cases which decreases the need for code duplication. In addition to these benefits, the jupyter notebook format also has educational benefits. With the ability to construct one notebook per use-case, and format documentation and useful information between the cells of code, the software designed becomes self-describing. It also allows users to execute the code one cell at a time, rather than the whole program at once. This allows users to effectively view the program step-by-step, and take in the documentation alongside the code, thus 195 furthering the ease of understanding what the code and the use-case are doing. This is hugely beneficial from a pedagogical standpoint.





**Table 1.** A list of the use-cases contained in the present version of the MDF Visualization toolkit.

| Use-case | Description | Data selection | References |
|---|---|---|---|
| time-series | Standard line-plot which shows observational and model data for a near surface variable over a selected range of dates. | The model data can be displayed either as a concatenated time-series, where only one day is used per forecast, or a stacked time-series where the full forecasts are shown overlapping each other. In the concatenated case the user can select which day for the forecast should be used for concatenation. | |
| scatter 1 | Shows model-observation agreement by plotting model data vs. the observation for near surface variables | Can show basic statistics such as a linear regression line and r-squared value. Or add a 1-to-1 line. | Day et al. (2023) |
| scatter 2 | 3 sub cases of functional diagrams of drag coefficient/ heat transfer. | The first case shows the relation between the friction velocity (ustar) vs. the wind speed, the second case shows the the relation between scaled sensible heat flux (sensible heat flux /wind-speed) and temperature gradient and the third the scaled water vapor flux and humidity gradient. | Tjernström (2005) Day et al. (2023) |
| contour plot | Two panel plot with contour plots for model and observation profile variables. | The model data is a concatenated time-series, user can select forecast day for concatenation. A slider exists to filter based on height. | Tjernström et al. (2021) |
| forecast error | Plots forecast error for near surface and profile variables. | The near surface case shows the distribution of forecast error defined as model minus observation vs forecast lead time The profile case shows time-height plots for the median forecast error vs. lead time. | Tjernström (2022) |
| Joint PDF | plots the joint PDFs for two variables with histogram/probability distribution | 3 cases, near surface error for specific humidity and temperature, near surface stratification and net surface long wave flux as well as observation and model for near surface variables. | |





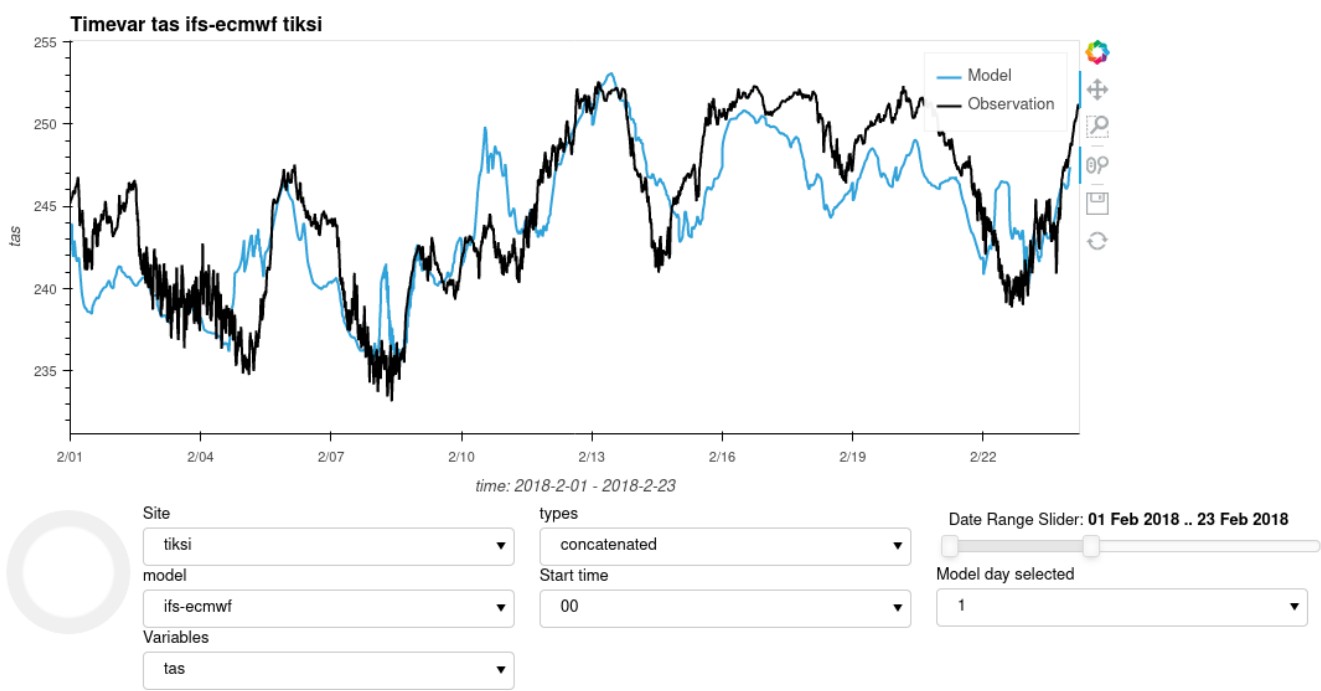

**Figure 7.** An example of a widget and plot produced by the visualization toolkit, showing the concatenated time-series use-case for the two meter temperature at Tiksi for the IFS model. The chosen forecast start time is 00, and concatenation is performed using the first day of the forecast for the dates Feb 1st 2018 to Feb 23rd 2018.

### 3.2.2 Web-interface

In order to be able to function as a web application alongside the YOPP database, the python plotting module Bokeh (Bokeh Team, 2023) was chosen to create the front-end for the visualization toolkit. In the cases where Bokeh plotting did not have the ability to create a plot, a combination of matplotlib (John Hunter and the Matplotlib development team, 2024) and "panes" from Pyhton library Panel (Holoviz Team, 2023) were used to make the matplotlib plot into a web application. Figure 7 shows an example of the web application created by the toolkit.

### 3.2.3 Widgets

The key point for the produced plots was to add high user customization, allowing the user a high degree of control over what is displayed. This was achieved through the python module panel (Holoviz Team, 2023), that can be used to create widgets, thus allowing the user the ability to customize the plots without having to provide their own input, Figure 7. The most commonly used widget across the use-cases is the drop-down menu, but sliders and buttons that toggle off and on plot features are also





used. Selections include the ability to choose which model and data site should be displayed, as well as which variable. Which dates, as well as forecast start time can also be selected. For cases that contain different plot options, or different analysis
options, these can also be selected, for example the time series use-case has one stacked plot version and one concatenated version of the NWP data.

### 3.2.4 Data handling and calculations

The back-end data reading, handling and processing functions exist as importable .py files and are written primarily using Xarray (Hoyer and Hamman, 2017), Pandas (The Pandas development team, 2020) and Numpy (Harris et al., 2020). The
current data fetching functionality works with urls to a thredds server, and can handle single files and concatenated or stacked datasets. The common data processing functionalities such as matching time-steps for plotting, calculating wind-speed and forecast error, all use numpy operations on pandas "dataframes", and exist as importable functions that can be used across the use-cases.

### 3.2.5 Testing

Community code, especially if it is designed to be built upon should meet certain standards in terms of code quality and maintainability. To this end three of the seven use-cases were tested, (Tjernström, 2022). The metric used to determine maintainability and code quality was defined as source code readability, which was quantified as a percentage of comments to lines of code as well as the readability of documentation using the Gunning's fog index (Aggarwal et al., 2002). In addition to this, visualization quality was judged, viewing the retention of context when switching color-schemes, as well as their interpretabil-
ity to the colorblind. The toolkit runtime was also considered as a metric of toolkit usability. The studied use-cases all passed the testing (Tjernström, 2022). The same standards set for the tested use-cases were maintained for the design of the other use-cases.

In addition to this testing the time-series use-case was demonstrated during the YOPP Final Summit (Wilson et al.), where participants could interact with the application and provide feedback. The application functioned well and the received feedback
referred to plot interpretability, such as the request for a more detailed legend, as well as the request for the use of long names for variables in the widgets.

## 4 Benefits of Open Tools to facilitate research

As discussed above, the existence of standardized and widely adopted formats can help keep data FAIR and facilitate research collaboration between communities. However, just the existence of a set standard does not directly impact the quality of the
files produced. The existence of toolkits to support a standardized format make sure that the standard is easy to use. This is done in three key ways: First, it makes the formats easy to apply, second, it safeguards the format interpretation, and last, it gives the format a platform on which a community of users and developers can be built. All this contributes to making the data creators take the extra step to make data available to the community and upholds the correct standard, which makes the data





produced more FAIR. In this way, both the standardized format and its toolkit serve to make data created in the MDF format
FAIR. It is easily findable and accessible, via its community and documentation, interoperable by format design and reusable
and reproducible by its metadata and version control.

However, simply providing a standardized format and openly available toolkit is not enough. The inclusion of good documentation also plays a key role in making the format and toolkits easy to use regardless of the experience level of the user. The goal is for the format to be easy to use even for novice users with little to no knowledge of its intricacies. This places
emphasis on straightforward functionality and well documented tools that allow users to start creating format adherent files without first having to immerse themselves in the format details. This also helps limit the amount of interpretation that is left to the user which reduces the risk of format divergent files. In this case the provided toolkits do most of the heavy lifting, and the documentation does the rest. The currently available documentation for the MDF format is contained within the H-K table (Hartten and Khalsa, 2022) as well as described in (Uttal et al., 2023).
With the basic format specification outlined efforts are being made to streamline the documentation to make it easier to digest for a first time user. To this end a repository has been created under the MDF-makers space on GitLab[2]. This page will contain both a broad overview, as well as more in depth documents. This will allow new users to, depending on engagement level, either just grasp the core contexts needed to get started, or more fully immerse themselves in the format. This documentation is intended to serve as a starting point for users who want to start applying the MDF format to their data and collects all the
information needed in one place to limit the amount of searching needed to find key format details.

## 4.1 Facilitating Format Growth Through Community Platforms and Engagement

The MDF format has been developed with expansion in mind and aims to make the process of adding variables, feature types and even new file types as simple as possible. The format itself provides the rules, naming conventions and structure that users can then apply and expand to suit their needs. In order to facilitate this process, and allow the official format specifications to
keep pace with the growth, a couple of efforts are being made.

The first key effort is to establish processes and policies for submission of variables and structures to the MDF format, as well as submissions of code to the toolkit suite. The code submission is handled via git merge requests and issues where people who have made their own edits or have suggestions can submit their code or ideas for the consideration of the core developers. The ability to easily suggest new variables or format changes is considered integral to MDF format growth, especially in
terms of expanding the specification to fit the needs of new communities. Variable additions and changes during YOPPsiteMIP were handled on a case by case basis, relying on key persons and email communication. The lack of an established process slowed down the H-K table development work, and the lack of traceable communication caused a lack of documentation for key changes. With this in mind, the submissions to the format specification and H-K table are set up via a git repository and using a submission template for the information required for variable consideration. Keeping both the documentation, code
and variable submission on the MDF-makers space on git makes the GitLab space a community platform where information,

---

[2]https://gitlab.com/mdf-makers





discussion and format news can be centralized and easy to access. Using GitLab as a central platform also allows the community to contribute to the code using pull requests and make suggestions using git issues.

To further expand the availability for collaboration and contact between the MDF makers and users a Discord[3] has been set up to serve as a second community platform alongside the gitlab page. This has been done with the aim of creating a place where the users of the format can communicate with each other and help each other as well as engage in discussion with the developers and format maintainers. This allows both groups to make use of each other's knowledge and experience, as well as giving the core MDF teams insight into the way the format is being applied. The Discord has been structured both for text communication in channels for asking questions, information channels where the developers can share current news, as well as voice channels where people can co-work or meetings and workshops can take place.

To maintain the MDF format and facilitate further growth, cross functional teams should be constructed that can process variable submission, update the H-K table, and maintain the MDF toolkit suite throughout its lifespan. The key to putting together these teams lies in the concept of cross functional, or x-functional, teams (Schwaber and Sutherland, 2011). This means that the teams include people with different backgrounds and expertise, meaning that data creators, and data users, observationalists and modellers as well as software developers collaborate on format growth. As new communities adopt the format they should also be included to facilitate growth that fits the needs of the new, larger community.

This is done to assure that format decisions are made with the different relevant disciplines in mind, which then simplifies the application of the format produced. Including developers from all user-groups will ensure that discipline specific use-cases and standards that need to be are taken into consideration at the first stage of development, thus limiting the number of iterations that are needed to reach a functional format change. This minimizes the risk that one change approved by a modeler will create problems for an observationalist using the format, or that a software development decision made by a software developer renders a tool useless to part of its broad user-base.

The application of cross functional teams is also relevant in the case of the toolkit suite. This helps to both identify new use-cases and functionalities but also to spread the knowledge of software architecture and maintenance over a larger group to avoid reliance on one core developer. This assures that the software will remain usable and maintained during its life-cycle even if the original developers do not remain involved. Thus the construction of cross-functional, dedicated teams for format and toolkit maintenance will be key to format survival and growth.

### 4.2 Reproducibility in Research and Open source toolkits

The production of standardized datasets with rich metadata is an important step towards reproducible research. However, while such a format does much to facilitate reproducibility, it is not entirely enough on its own. Making use of an openly available toolkit to fit the standard also facilitates reproducibility, as it ensures that the processing of the data remains constant across all files being created. It also, if versioned and referenced properly, gives the user the ability to recreate earlier versions of the files from raw data. It provides a way to track the whole process, from the raw data, through the data processing, to the creation

---

[3]https://discord.gg/nYhNYu8S55



of a file. Hence, providing a standardized format, with in-depth metadata, including versioning, and an openly available well documented toolkit makes the data produced and the research based on it more reproducible.

305 To this end the MDF toolkit suite uses Git for version control, and Zenodo to be able to cite the version used in the files that are being created. This as well as the metadata rich nature of the MDF format, alongside specific attention paid to versioning (Uttal et al., 2023) and the availability of version controlled toolkits makes the MDF format especially reproducible.

## 5  Conclusions and Future work

The Merged Data File (MDF) format was created to increase the structure and usability of observational and model data with 310 the intent to facilitate process level model evaluation and development (Uttal et al., 2023). The format and toolkits outlined in this paper have contributed to the over all goals set for the Year of Polar Prediction project (Jung et al., 2024).

The MDF is based on the netCDF format, and outlines merged model data files (MMDFs) and merged observation data files (MODFs) that both rely on a common set of variable naming conventions, building on CF conventions as outlined in the Hartten-Khalsa table (Hartten and Khalsa, 2022). The usage of MDF files in the Year of Polar prediction project has aided the 315 ability to easily perform model evaluation as is outlined in (Day et al., 2023) and (Mariani et al., 2023). With the basic format in place, as described in (Uttal et al., 2023), current efforts are focused on community building. To facilitate this, as well as aiding the existing community, a suite of tools was designed to help creating and visualizing MDF files.

Future work in relation to these toolkits is planned. For instance, the MDF creation toolkit was initially designed to handle observational data hence work is being done to expand its functionality to be more general in which data types it can handle. 320 There is also room to add automated quality control and other features to broaden the creation toolkit from its initial specific use-case to a more generic data processing framework. The basic functionality of the visualization toolkit is in place, but there is ample room for growth based on the needs of the community. The web-application formats for the use-cases are being finalized and other functionality such as reading files locally is planned. There are also intentions to incorporate the available use-cases into the YOPP data portal. In addition to this there is room to expand in terms of available use-cases, but the current 325 focus of the development team lies on evolving and spreading the MDF format as well as completing the basic functionalities of the creation toolkit. Updates to the format, as well as more in depth documentation for the H-K table and toolkits can be found on the MDF makers space on Gitlab [4]. The ultimate goal is to grow and expand the MDF community both in regards to a polar context, including more sites and models, as well as in regards to collaborating with other communities.

---

[4]https://gitlab.com/mdf-makers



*Code and data availability.* The MDF toolkit is freely available for use as an open source software. It is licensed under version three of
GNU General Public License, a copyleft license allowing all users to run, study, share, modify, and redistribute the MDF toolkit code.
The team follows an open development model and contributions to the code base can be submitted at the Gitlab hosted code repository
(https://gitlab.com/mdf-makers/mdf-toolkit). Released packages may be referenced on Zenodo (https://doi.org/10.5281/zenodo.7814813).

The visualization toolkit is freely available for use as an open source software made available via GitLab. The team follows an open
source development model and contributions to the code-base can be submitted at the Gitlab hosted code repository (https://gitlab.com/mdf-
makers/mdf-visualisation-toolkit) The code may be referenced on Zenodo (https://doi.org/10.5281/zenodo.12664308).

The data files used and developed by the toolkits in this paper can be found on the YOPP data portal (https://yopp.met.no/), and are
discussed in (Day et al., 2023; Mariani et al., 2023).

*Author contributions.* The original idea and outline of the creation toolkit was designed by MG and JH. The visualization toolkit was
developed by JT under the guidance of SJK, JD and LF. The first draft of this paper was written by JT and GS. All authors contributed to the
writing and editing of the paper as well as the design of the toolkits.

*Competing interests.* The contact author has declared that none of the authors have any competing interests.

*Acknowledgements.* This is a contribution to the Year of Polar Prediction (YOPP), a flagship activity of the Polar Prediction Project (PPP)
initiated by the World Weather Research Programme (WWRP) of the World Meteorological Organization (WMO). We acknowledge the
WMO WWRP for its role in coordinating this international research activity. Many thanks to everyone involved with YOPPsiteMIP, without
your guidance and stimulating discussions this would not have been possible. Thank you to MET Norway for hosting the YOPP Data Portal
and Massimo Di Stefano for your guidance on the visualization toolkit implementation.

*Financial support.* Jonathan Day was supported by the INTERACT III project funded by the European Union (grant agreement no. 871120).
Leslie M. Hartten was supported in part by NOAA cooperative agreements (grant nos. NA17OAR4320101 and NA22OAR4320151). Michael
Gallagher and Matthew D. Shupe acknowledge funding support from the US Department of Energy Atmospheric System Research Program
(DE-SC0019251) and a NOAA cooperative agreement (NA22OAR4320151).



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
