# Peer review of "Accelerating research through community open source software for a standardized file format to improve process representation in numerical weather prediction models"

_EGUsphere, 2024_

## Referee Comment (RC2)

**Review of "Accelerating research through community open source software for a standardized file format to improve process representation in numerical weather prediction models"**

**General comments**

The manuscript showcases a toolkit to standardize observational data in a way that would make it easier for model physics developers to utilize observational data from non-standard sources in their work. The manuscript presents two open source Python packages, one dealing with creation of the standardized data ("Merged Data Format") and one with visualization of this data.

The overarching goal presented in the manuscript is commendable – it would indeed be very useful for model developers to be able to verify their model forecasts more easily against different observational datasets such as those made during YOPP as discussed in the manuscript. However, I feel the manuscript in its current state is nowhere near publishable: it currently reads to me more like an internal project documentation rather than a scientific article presenting an innovative new observation handling toolkit (that it should do).

**Major issues**

1) What is lacking in my opinion is a structure that would allow the reader to understand the logic and actual benefits of using the MDF format. One way of improving on this would be to showcase in detail what the code does through presenting a clear example of the workflow. What I mean by this is to include a detailed example of what the MDF toolkit actually does, an example:

   i.   Present some observational datasets that do not follow the same (meta) data format (something similar to ncdump -h output). It would be nice to see how a) two different similar observations (e.g. tower data) are harmonized by MDF, b) and how dealing with e.g. sounding data differs to that of a point observation data.
   ii.  Present in detail how MDF toolkit logic handles reformatting these different datasets.
   iii. Present how the (meta) data format looks like after MDF toolkit handling.
   iv.  Repeat i)-iii) for 1-2 model forecasts.
   v.   Present how these observations and forecasts in MDF format would be visualized with the MDF visualization tool (show examples of code call and plots generated).

2) netCDF is referenced as the basis for the MDF format, but the manuscript does not state whether the MDF format is actually netCDF with standardized meta data, coordinates etc., or a completely different data format that won't be readable through netCDF libraries and applications (e.g. ncview, GRIB-API/ecCodes). Please clarify.

3) I do not know if this is an issue caused by the language used or do the authors actually share this line of thought, but currently, especially the introduction, presents **all** observations to be problematic for modellers. There should be a clear distinction between observational campaigns and non-operational observational datasets that are (or might be) problematic, and long-running operationally handled observations that already conform to a standard (operational soundings, SYNOP, METAR, …) and are used daily by operational NWP models. Please make this distinction clear.

3) The grammar of the manuscript needs considerable work (punctuation rules etc.).

**Minor issue**

1)  Could the authors expand on why comparing multiple models would be an issue of (P2 L46)?

2) Again, please check your language, constructs like "process based NWP" should not be in (if a NWP model does not include physical parametrizations, i.e. "processes", they are dynamical models).

3) Table 1: is observational error included and is plotting it a part of the visualization toolbox?

4) Why are you only testing 3 out of your 7 test-cases against standards? Shouldn't you aim for all pieces of your program to meet these standards?

---

## Author Comment (AC1)

We thank the reviewers for their comments. The comments are repeated below and answers are *in italic*.

RC1: This study accelerates the research on standardized file formats through community open-source software to improve process representation in numerical weather prediction models. This topic helps address the challenges posed by climate change and promotes collaboration and innovation within the scientific community. However, the paper has several significant deficiencies: (1) The structure of the paper is flawed, making it very difficult to read. (2) The research content is weak and insufficient to support a publishable academic paper in GMD. (3) Aside from introducing the usage and applications of an open-source plugin, the paper lacks adequate introduction and analysis of the plugin's model and methodological advancements. In summary, this paper has considerable distance to cover before it can be considered a publishable academic work.

*We appreciate the honest comments about the topic of the manuscript as well as its deficiencies. We will restructure the manuscript, enhancing the content with more examples of issues using existing data sources and benefits of the file format and the toolkits. We will also present the software construction and use. We feel encouraged to continue developing the work and this manuscript.*

RC2:

**General comments**

The manuscript showcases a toolkit to standardize observational data in a way that would make it easier for model physics developers to utilize observational data from non-standard sources in their work. The manuscript presents two open source Python packages, one dealing with creation of the standardized data ("Merged Data Format") and one with visualization of this data. The overarching goal presented in the manuscript is commendable – it would indeed be very useful for model developers to be able to verify their model forecasts more easily against different observational datasets such as those made during YOPP as discussed in the manuscript. However, I feel the manuscript in its current state is nowhere near publishable: it currently reads to me more like an internal project documentation rather than a scientific article presenting an innovative new observation handling toolkit (that it should do).

*We are pleased that the reviewer recognizes the value of the tools that we have developed and are discussing in the manuscript. We agree with the reviewer that we have not been able to present the material sufficiently to get our message across. The cross-disciplinary nature of this research may be a contributing factor as publication traditions vary. We are committed to amend these issues and are very grateful for the suggestions.*

**Major issues**

1) What is lacking in my opinion is a structure that would allow the reader to understand the logic and actual benefits of using the MDF format. One way of improving on this would be to showcase in detail what the code does through presenting a clear example of the workflow. What I mean by this is to include a detailed example of what the MDF toolkit actually does, an example:

i. Present some observational datasets that do not follow the same (meta) data format (something similar to ncdump -h output). It would be nice to see how a) two different similar observations (e.g. tower data) are harmonized by MDF, b) and how dealing with e.g. sounding data differs to that of a point observation data.

ii. Present in detail how MDF toolkit logic handles reformatting these different datasets.

iii. Present how the (meta) data format looks like after MDF toolkit handling.

iv. Repeat i)-iii) for 1-2 model forecasts.

v. Present how these observations and forecasts in MDF format would be visualized with the

MDF visualization tool (show examples of code call and plots generated).

*We appreciate the suggestion to include more detailed examples of the workflow to make the issues and the solution more concrete. We will rewrite Section 3 "The Merged Data File toolkit suite" to include explicit information on the intended workflow with examples from various data sources. This aims to better illustrate the issues of the existing working environment and the benefits of the MDF format and the toolkits. This will primarily be done by restructuring and rewriting Section 3 to place a greater focus on the workflow rather than the individual toolkits with the additions mentioned above.*

*We will also include specific examples of the variations of datafiles, both observational and models, that can easily be accessed in open source "standardized" databases, to motivate the need for MDF and how the toolkits are helpful. We will also include examples that compare what information that can be deduced with MDF files in contrast to more standard observations and model output to better illustrate the need. The examples may include temporal and spatial resolution and also the impact of interpolating or not, with the topic of process based evaluation in mind.*

2) netCDF is referenced as the basis for the MDF format, but the manuscript does not state whether the MDF format is actually netCDF with standardized meta data, coordinates etc., or a completely different data format that won't be readable through netCDF libraries and applications (e.g. ncview, GRIB-API/ecCodes). Please clarify.

*We will clarify the description of the MDF format with respect to these comments. The files are netCDF files with standardized meta data and structure and thus work with common netCDF libraries. More details on the format specifics are found in Uttal et al., 2023.*

3) I do not know if this is an issue caused by the language used or do the authors actually share this line of thought, but currently, especially the introduction, presents all observations to be problematic for modellers. There should be a clear distinction between observational campaigns and non-operational observational datasets that are (or might be) problematic, and long-running operationally handled observations that already conform to a standard (operational soundings, SYNOP, METAR, …) and are used daily by operational NWP models. Please make this distinction clear.

*The overarching aim of the MDF file format is to improve the workflow for modelers that are trying to improve numerical models through better process description. We will carefully rewrite and add material to make this clear and to make the distinction between operational data and its handling. Although the operational data is easy to access within NWP institutes, this is not the case for university scientists or students.*

3) The grammar of the manuscript needs considerable work (punctuation rules etc.).

*We will carefully take these comments under consideration.*